# Oxo dicopper anchored on carbon nitride for selective oxidation of methane

Pengfei Xie [1,2,7 ✉], Jing Ding[1,3,7], Zihao Yao [4,7], Tiancheng Pu [1,7], Peng Zhang [5], Zhennan Huang[6], Canhui Wang[1], Junlei Zhang[1], Noah Zecher-Freeman[1], Han Zong[1], Dashui Yuan[3], Shengwei Deng[4], Reza Shahbazian-Yassar [6] & Chao Wang [1 ✉]

Selective conversion of methane ($CH_4$) into value-added chemicals represents a grand challenge for the efficient utilization of rising hydrocarbon sources. We report here dimeric copper centers supported on graphitic carbon nitride (denoted as $Cu_2@C_3N_4$) as advanced catalysts for $CH_4$ partial oxidation. The copper-dimer catalysts demonstrate high selectivity for partial oxidation of methane under both thermo- and photocatalytic reaction conditions, with hydrogen peroxide ($H_2O_2$) and oxygen ($O_2$) being used as the oxidizer, respectively. In particular, the photocatalytic oxidation of $CH_4$ with $O_2$ achieves >10% conversion, and >98% selectivity toward methyl oxygenates and a mass-specific activity of 1399.3 mmol g $Cu^{-1}h^{-1}$. Mechanistic studies reveal that the high reactivity of $Cu_2@C_3N_4$ can be ascribed to symphonic mechanisms among the bridging oxygen, the two copper sites and the semiconducting $C_3N_4$ substrate, which do not only facilitate the heterolytic scission of C-H bond, but also promotes $H_2O_2$ and $O_2$ activation in thermo- and photocatalysis, respectively.

[1] Department of Chemical and Biomolecular Engineering, Johns Hopkins University, Baltimore, MD 21218, USA. [2] College of Chemical and Biological Engineering, Zhejiang University, Hangzhou 310027, China. [3] State Key Laboratory of Materials-Oriented Chemical Engineering, College of Chemical Engineering, Nanjing Tech University, Nanjing 211816, P.R. China. [4] Institute of Industrial Catalysis, College of Chemical Engineering, Zhejiang University of Technology, Hangzhou 310014, China. [5] School of Materials Science and Engineering, Zhengzhou University, Zhengzhou 450001, China. [6] Department of Mechanical and Industrial Engineering, University of Illinois, Chicago, IL 60607, USA. [7] These authors contributed equally: Pengfei Xie, Jing Ding, Zihao Yao, Tiancheng Pu. ✉email: pfxie@zju.edu.cn; chaowang@jhu.edu

Selective conversion of methane to liquid hydrocarbons represents a promising approach toward efficient utilization of natural gas[1]. The present industrial route for such conversions relies on a two-step process by first reforming methane to generate synthesis gas (CO and $H_2$) at elevated temperatures (>500 °C), and then reacting CO with $H_2$ to form methanol or other liquid products[2,3]. However, this process is energy-intensive and economically nonviable for distributed sources such as flare gas[4]. More robust technologies toward direct conversion of methane into condensed energy carriers are thus demanded to facilitate transportation and storage[5].

Direct, partial oxidation of methane to methyl oxygenates has received intensive attention in recent years[6,7]. The studies on early days use transition copper exchanged zeolites to catalyze the reaction between $CH_4$ and $O_2$, and employ a two-step chemical looping process to subsequentially activate $O_2$ and desorb the products[8–10]. Despite the achievement of high selectivities, these reactions are still suffering from the low $CH_4$ conversions (typically < 0.03%) and reaction rates (<30 $\mu$mol $g_{cata}^{-1}$ $h^{-1}$)[11–13]. Later on, partial oxidation of methane in a single step has been demonstrated by using non-$O_2$ oxidizers such as oleum[14]), selenic acid[15]) and $H_2O_2$[16–18], but the cost associated with these oxidizing agents holds back their practical implementations[19]. More recent efforts have thus turned into in-situ generation of $H_2O_2$ from $O_2$ by using selective oxygen reduction catalysts such as Au-Pd containing zeolites[20]. Alternatively, photoexcitation using visible light is proposed to be advantageous with near-room temperature activation of $CH_4$, mitigating the concern of over oxidation to form $CO_2$ upon heating[21,22]. But the reported photocatalytic oxidation of methane is still limited by relatively low methane conversions (<1%) and productivities (0.001~150 mmol $g_{cata}^{-1}$ $h^{-1}$)[22], as the commonly used photocatalysts have quite large bandgaps (e.g., ~3.2 eV for $TiO_2$[23] and ~3.4 eV for ZnO)[21] and may only activate methane via the Fenton or homolytic mechanisms that have relatively sluggish kinetics[24]. In this aspect, graphitic carbon nitride (g-$C_3N_4$) represents a promising photocatalytic substrate with a modest band gap in the range of 2.7-2.9 eV[25–27]. Its abundant nitrogen sites have been shown in many reports to be capable of anchor atomically dispersed transition metal sites[28]. It thus becomes interesting to investigate the potential coordination of active Cu sites on g-$C_3N_4$ and examine their synergies in partial oxidation of methane.

Here we report on $Cu_2@C_3N_4$ as highly efficient catalysts for partial oxidation of methane. The dimeric copper catalysts were synthesized by supporting an (oxalato)(bipyridine)copper(II) complex, $[Cu_2(bpy)_2(\mu\text{-ox})]Cl_2$, on g-$C_3N_4$ and then applying a mild thermal treatment in air (Fig. 1a). The derived catalysts contained dicopper-oxo centers anchoring on g-$C_3N_4$ via four Cu-N bonds (two for each copper atom), as characterized by using STEM, XPS and XAS, and also confirmed with atomistic simulations. The obtained copper-dimer catalysts were first evaluated for thermal oxidation of methane using $H_2O_2$ as the oxidizer, and then further applied for photocatalytic oxidation of methane with $O_2$. Mechanisms governing the observed catalytic enhancements toward selective oxidation of methane were interpreted via combining computational simulation of the reaction pathways, spin-trapping EPR analysis of possible radical intermediates, and in situ XPS measurements under light irradiation.

## Results and discussion
### Synthesis and characterization of $Cu_2@C_3N_4$.
The copper-dimer precursor $[Cu_2(bpy)_2(\mu\text{-ox})]Cl_2$ was first prepared by a complexation reaction of copper chloride ($CuCl_2$), 2,2,-bipyridine and oxalic acid[29]. The g-$C_3N_4$ substrate was grown by calcination

of urea at 550 °C[30]. $Cu_2@C_3N_4$ catalysts were synthesized by self-assembly of the dimeric copper complex on g-$C_3N_4$[31] and then treating the mixture in air at 250 °C to immobilize the copper species (Fig. 1a). The loading of Cu was determined to be 0.35 wt% by using inductively coupled plasma mass spectrometry (ICP-MS).

The complexation of pyridine, $Cu^{2+}$ and oxalate ($C_2O_4^{2-}$) to form an organometallic compound was confirmed by using Fourier transform infrared spectroscopy (FTIR). The hydroxyl (O-H) and carbonyl (C = O) stretching features around 3450 and 1670 $cm^{-1}$, respectively, associated with oxalic acid disappeared after the reaction. This was accompanied with the blue shift of the characteristic band (attributed to the asymmetric stretching of the pyridyl ring) of 2,2′-pyridine at ca. 1580 $cm^{-1}$ to ca. 1650 $cm^{-1}$, a consequence of its chelation with $Cu^{2+}$ (Fig. 1b)[32]. Correspondingly, the $d$-$d$ transition of $Cu^{2+}$ at 650–700 nm had a blue shift of 48 nm in the ultraviolet-visible diffuse reflectance spectroscopy (UV-vis DRS) patterns (Fig. 1c). The FTIR spectra of g-$C_3N_4$ and $Cu_2@C_3N_4$ exhibited the stretching vibration modes characteristic of the −NH group around 3180 $cm^{-1}$, C-N heterocycles in the wavelength range of 1100–1650 $cm^{-1}$ and the breathing mode of tri-$s$-triazine units at 810 $cm^{-1}$ (Fig. 1d). Comparing to $[Cu_2(bpy)_2(\mu\text{-ox})]Cl_2$, $Cu_2@C_3N_4$ presented no more infrared features associated with the dimeric copper complex, indicating the complete removal of organic ligands during the immobilization process. This was further confirmed with thermogravimetric analysis (TGA) (Supplementary Fig. 1). The $[Cu_2(bpy)_2(\mu\text{-ox})]$ $Cl_2$/g-$C_3N_4$ mixture lost ~6% of its initial weight upon annealing in air at up to 250 °C, which is close to the expectation estimated based on the ligand content (~80 wt%) of $[Cu_2(bpy)_2(\mu\text{-ox})]Cl_2$ and its ratio relative to the carbon nitride substrate (~8%) used in the synthesis. X-ray diffraction (XRD) patterns collected for the $Cu_2@C_3N_4$ catalysts only show the (001) and (002) peaks associated with g-$C_3N_4$, with the absence of copper metal or oxide features indicating the highly dispersed nature of copper species (Supplementary Fig. 2). Atomic structures of the dimeric copper moieties were resolved by using aberration correction high angle annular dark field scanning transmission electron microscopy (HAADF-STEM) imaging (Fig. 1e-g). The collected STEM images exhibit a large number of adjacent, paired bright dots (labeled with red circles, <0.35 nm in size for each dot) distributed on a substrate of lower contrasts. These small bright dots can be attributed to atomically dispersed Cu considering their much higher Z contrast (M = 65 for Cu) than $C_3N_4$ (M = 12 or 14). Line-profile scanning for ~100 pairs of such bright dots give an average distance of 2.8 (±0.2 Å) (Fig. 1h). This is much shorter than the value (5.2 Å) for the two copper atoms within $[Cu_2(bpy)_2(\mu\text{-ox})]Cl_2$, again confirming the reconstruction and condensation of the copper-dimer moieties as a result of the removal of organic ligands in the synthesis.

Chemical nature of the Cu dimers in $Cu_2@C_3N_4$ was probed by using X-ray photoelectron spectroscopy (XPS) and X-ray absorption spectroscopy (XAS). The N $1s$ XPS spectrum exhibits a broad feature with the binding energy ranging from 397 to 408 eV (Fig. 2a). This feature can be deconvoluted into four peaks centered at 398.6, 399.4, 401.0, and 404.5 eV, which can be assigned to pyridinic (C − N = C), tertiary (−N<), pyrrolic (−NH) nitrogen, and π-π* transition of C = N or uncondensed terminal amine groups in g-$C_3N_4$, respectively[33,34]. The Cu $2p$ spectrum shows two peaks at 932.5 eV and 952.3 eV, which are characteristic of Cu(I) or $Cu^0$ (Supplementary Fig. 3)[35]. However, the XPS analysis (as well as the corresponding Auger electron spectrum, AES) was unable to explicitly determine the oxidation state of Cu due to the reduced signal-to-noise ratio associated with the low copper content in the catalysts. The copper oxidation state in $Cu_2@C_3N_4$ was better resolved by using

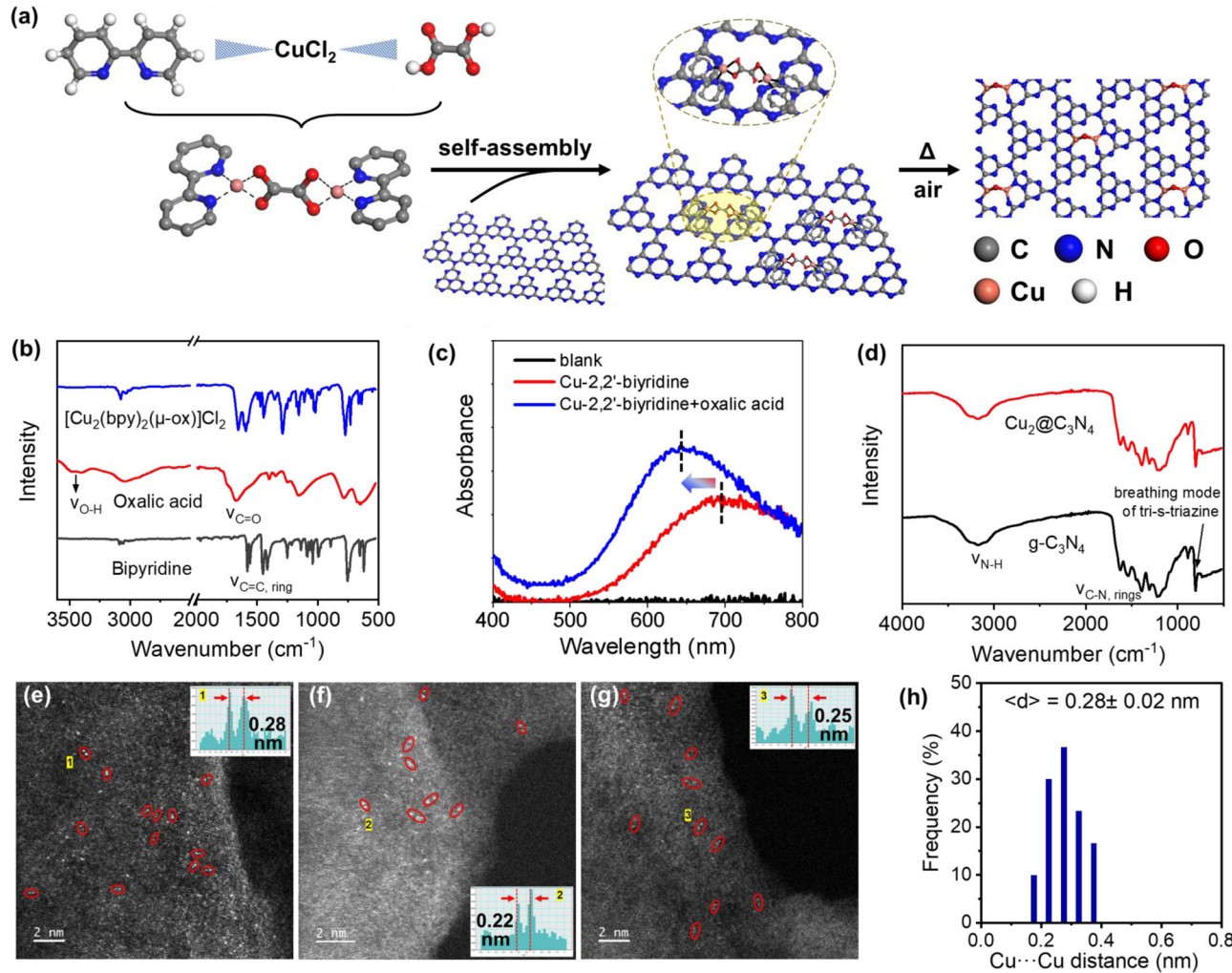

**Fig. 1 Synthesis of the Cu$_2$@C$_3$N$_4$ catalysts. a** Scheme illustration of synthetic route. **b, c** Characterization of the Cu-dimer precursor, [Cu$_2$(bpy)$_2$(μ-ox)] Cl$_2$ complex, using FTIR (**b**) and UV-vis DRS (**c**). **d** Comparison of FTIR spectra for Cu$_2$@C$_3$N$_4$ and g-C$_3$N$_4$. **e–g** Representative HAADF-STEM images of Cu$_2$@C$_3$N$_4$, with the insets showing line-scanning intensity profiles of Cu dimers. **h** Statistical distribution of the Cu-Cu distance in the Cu dimers derived from the STEM images.

X-ray Absorption Near Edge Spectroscopy (XANES) (Fig. 2b). The Cu K edge spectrum exhibits a pre-edge transition at 8984 eV, which falls between the peaks associated with Cu$_2$O (8983 eV) and CuO (8986 eV). This indicates an intermediate oxidation state between +1 and +2 for Cu in Cu$_2$@C$_3$N$_4$. Noticeably, our results do not support the picture with a mixture of Cu(I) and Cu(II), as the *1 s → 3d* transition at 8977 eV, a feature characteristic of Cu$^{2+}$ (as shown for the references CuO and copper tetraphenylporphyrin (Cu-TPP) in Supplementary Fig. 4)[35], is absent in the spectrum of Cu$_2$@C$_3$N$_4$. The partial oxidation state (between +1 and +2) of Cu within the copper dimers supported on g-C$_3$N$_4$ can be viewed as a result of the semiconducting nature of the substrate. This is distinguished from the case for extensively studied copper exchanged zeolites, in which the dicopper-oxo centers ([Cu-O-Cu]$^{2+}$) anchor on the Al sites with localized negative charges and have an oxidation state of +2 for both Cu atoms[9,36,37].

The atomic structure of the Cu dimers was resolved by combining extended X-ray absorption fine structure (EXAFS) analysis and atomistic modeling based on DFT calculations. Figure 2c compares the k2-weighted Cu K edge EXAFS spectra for Cu$_2$@C$_3$N$_4$, Cu foil, Cu$_2$O, CuO, and Cu-TPP (with single-atom Cu$^{2+}$ coordinating to four pyrrolic N, Supplementary Fig. 4). The Cu$_2$@C$_3$N$_4$ catalyst

exhibits first-shell scattering at 1.62 Å in R space (prior to phase correction), which is proximate to the values, 1.59 and 1.55 Å, found for Cu-TPP and CuO, respectively. This is distinct from the cases for Cu$_2$O and Cu, the first-shell scattering of which locates at 1.50 and 2.30 Å, respectively. From these observations, we tentatively assign the primary scattering pair at 1.62 Å in the R-space spectrum of Cu$_2$@C$_3$N$_4$ to be Cu-N or Cu-O bonding. To fit the EXAFS spectrum, a total of 9 possible Cu-dimer configurations was postulated, based on which DFT calculations were performed to relax the structures and determine bonding distances and angles (Supplementary Fig. 5 and Table 1). Considering the presence of minor Cu monomers observed in the STEM images (Fig. 1e–g), we have applied a linear combination of Cu monomers and dimers to fit the EXAFS spectrum. Various possible Cu-dimer configurations have been considered, with the corresponding EXAFS spectra compared to the experimental results to identify the best fit (Fig. 2d, e; also see Supplementary Figs. 6–7). It is estimated that 72.4% of the Cu atoms are in the dimeric configuration, close to the value (~70%) derived from statistical analysis of the STEM images. The determined copper-dimer structure comprises two Cu atoms bridged by an O atom, with the Cu-O bonds having lengths of 1.76 Å and 1.79 Å and an included angle (∠Cu-O-Cu) of 99.6° (Fig. 2f, Table 1). Each Cu atom is coordinated to two N atoms on

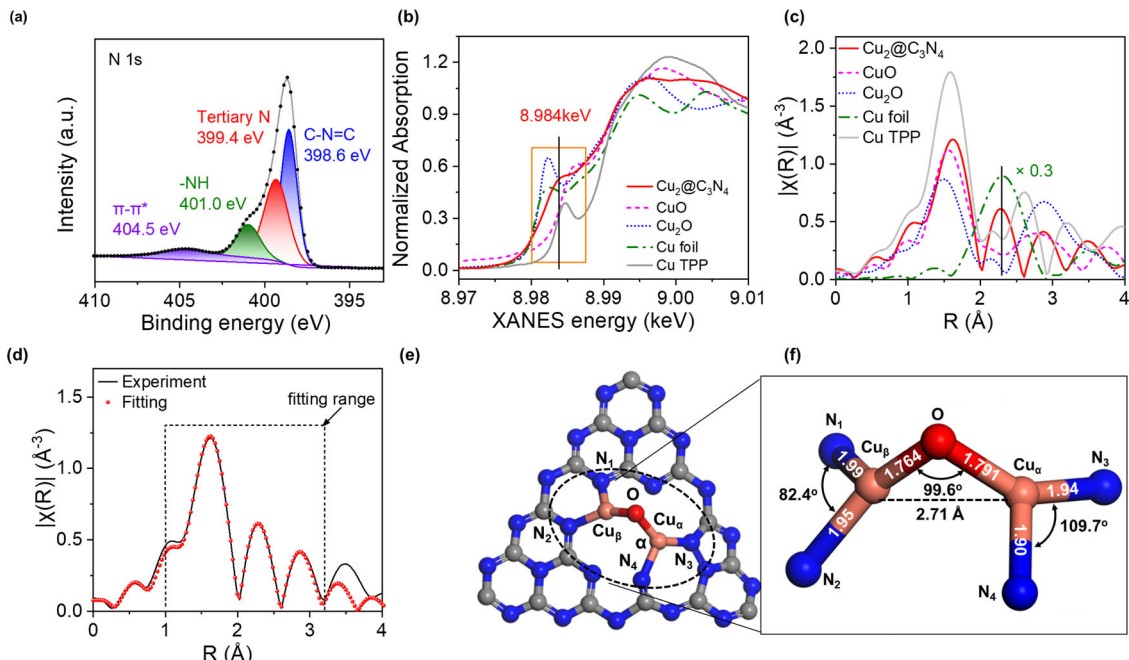

**Fig. 2 Characterization of the Cu$_2$@C$_3$N$_4$ catalysts. a** XPS spectrum at the N *1s* edge and the corresponding deconvolution. **b** XANES spectra and (**c**) $k^2$-weighted EXAFS spectra at the Cu K edge, with Cu foil, Cu$_2$O, CuO, and Cu-TPP (one Cu coordinated with for N atoms) as the reference. **d** Fitting of the EXAFS spectrum with consideration of both monomeric and dimeric Cu sites. **e** The simulated structure model of dicopper-oxo center. **f** Geometric parameters of the dicopper-oxo center determined for Cu$_2$@C$_3$N$_4$.

**Table 1 Structural parameters being used to fit the EXAFS spectrum for Cu$_2$@C$_3$N$_4$ with consideration of both monomeric and dimeric Cu sites.**

| Scattering path | CN | Distance (Å) | σ2(Å$^2$) | R-factor |
|---|---|---|---|---|
| Cu-O | 1.25 ± 0.20 | 1.77 ± 0.01 | 0.0056 ± 0.0006 | 0.009 ± 0.001 |
| Cu-N (Cu dimer) | 1.84 ± 0.29 | 1.99 ± 0.01 | | |
| Cu-N (Cu monomer) | 2.42 ± 0.32 | 1.98 ± 0.01 | | |
| Cu-Cu | 0.88 ± 0.15 | 2.71 ± 0.01 | 0.0038 ± 0.0004 | |

the C$_3$N$_4$ framework, with the bonding distance varying from 1.90 to 1.99 Å and the bonding angle (∠N-Cu-N) being 82° for Cu$_\alpha$ and 110° for Cu$_\beta$ (Fig. 2f). Noticeably, the identified configuration with best fitting to the EXAFS spectrum also has the lowest (most negative) formation energy among the various configurations, in line with the expectation for stable atomic structures in the real catalysts (Supplementary Table 1). Our combined EXAFS analysis and DFT calculations resolved the Cu-Cu distance in the Cu dimers to be ~2.71 Å (Table 1), which is in agreement with the average Cu-Cu distance measured from the STEM images (Fig. 1h). It is noted that this value is much smaller than the Cu-Cu distance (4.10 Å) associated with the dicopper-oxo center ([Cu-O-Cu]$^{2+}$) in Cu-ZSM-5[37]. Furthermore, Bader charge analysis based on DFT calculation shows that the Cu atoms in the Cu dimers have oxidation states of +1.63 and +1.72 (Supplementary Fig. 7), resembling the results derived from XANES spectra (see the above discussion for Fig. 2b).

From the above discussion, we can see that the dimeric copper centers in Cu$_2$@C$_3$N$_4$ have distinct atomic structures and electronic chemical properties form their counterparts confined in zeolites. We hypothesize that their non-integer oxidation state (intermediate between +1 and +2) and reduced cluster size (smaller Cu-Cu distance as compared to Cu-ZSM-5) would lend them exquisite catalytic performance for selective oxidation of methane[36].

**Thermocatalytic oxidation of methane with H$_2$O$_2$.** The Cu$_2$@C$_3$N$_4$ catalysts were first evaluated for the thermocatalytic

oxidation of CH$_4$ (Supplementary Figs. 8–12). This was conducted using a continuous stirred-tank reactor (CSTR) filled with 0.2 mM of H$_2$O$_2$ and 0.1 MPa of CH$_4$ (see Experimental Methods in the Supplementary Materials). Methyl oxygenates (CH$_3$OH and CH$_3$OOH) were found to be the primary products, with the yield achieving 0.14% within 30 min of reaction at 30 °C (Fig. 3a). As previously reported, the generated CH$_3$OOH can be facilely reduced to CH$_3$OH under ambient conditions (Supplementary Fig. 12)[19,38]. The yield of methyl oxygenates increased to 0.37% at 70 °C, corresponding to the increase of productivity from 51.6 to 129.7 mmol g$_{Cu}^{-1}$ h$^{-1}$. Albeit the increase of reaction rate, the rise of reaction temperature is accompanied with the increase of CO$_2$ selectivity from 0.8% at 30 °C to 5.0% at 70 °C (Supplementary Fig. 13). Similarly, elongated operations also led to the yield of more CO$_2$ (Supplementary Fig. 13). The cyclability tests showed that the Cu$_2$@C$_3$N$_4$ catalyst was stable throughout the methane oxidation reaction with H$_2$O$_2$. In five consecutive measurements by refilling CH$_4$ and H$_2$O$_2$, the Cu$_2$@C$_3$N$_4$ catalyst exhibited indiscernible change in reactivity and product distribution, with the productivity of methyl oxygenates found to be consistent at ~70 mmol g$_{Cu}^{-1}$ h$^{-1}$ at 50 °C (Supplementary Fig. 14). Furthermore, the atomic structure of dicopper-oxo centers was confirmed to remain intact after reaction by performing HADDF-STEM imaging and EXAFS analysis on the spent Cu$_2$@C$_3$N$_4$ catalyst after 6 h of reaction (Supplementary Fig. 15 and Supplementary Table 2).

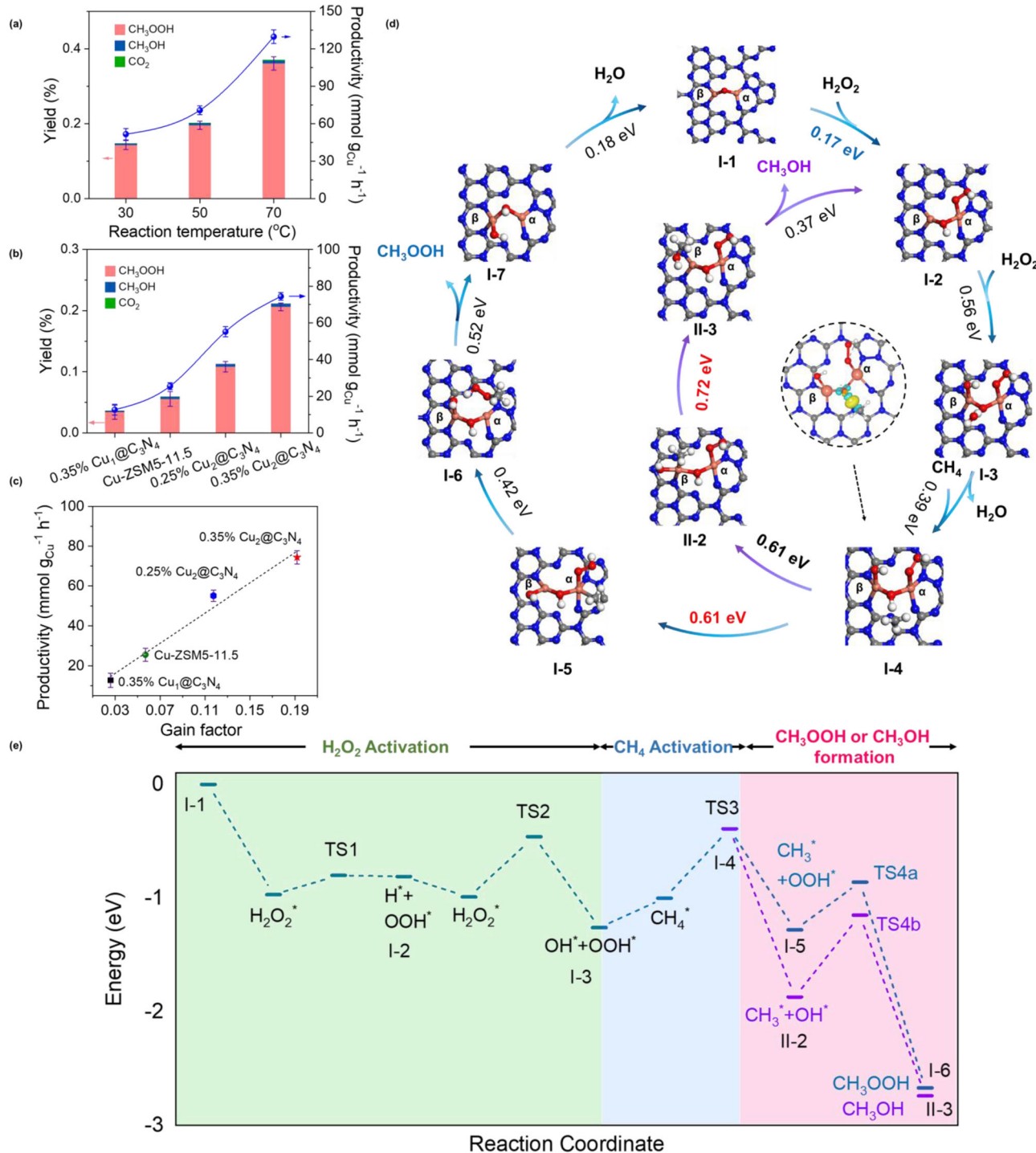

**Fig. 3 Thermo-catalytic oxidation of CH$_4$ with H$_2$O$_2$. a** Yields and productivity of methyl oxygenates at different reaction temperatures. **b** Comparisons of product yields and productivity over different catalysts. **c** Correlation between productivity of methyl oxygenates and gain factor for different catalysts. **d** Simulated pathways for the reaction between CH$_4$ with H$_2$O$_2$ on the Cu$_2$@C$_3$N$_4$ catalysts, with the middle inset illustrating the electron distribution of the CH$_4$ molecule being activated on the bridging oxygen site. Energy barriers are also given for the associated molecular transformations. **e** The DFT calculated free energy diagram for the Cu$_2$@C$_3$N$_4$-catalyzed partial oxidation of CH$_4$ with H$_2$O$_2$. Three stages consisting of H$_2$O$_2$ activation, CH$_4$ activation and methyl oxygenates formation are distinguished with different colors. The error bars presented in (**a–c**) indicate the statistical distribution derived from three independent measurements.

Considering that the bare g-C$_3$N$_4$ substrate is inactive for CH$_4$ oxidation (Supplementary Fig. 16), the dicopper-oxo centers can be identified as the active sites in Cu$_2$@C$_3$N$_4$. Considering that monomeric Cu in copper exchanged zeolites[10,39,40] or metal−organic frameworks[41] has also been discussed to be active for CH$_4$ oxidation, we have performed comparative studies on a single-atom control (Cu$_1$@C$_3$N$_4$) using the same g-C$_3$N$_4$ substrate. This catalyst was prepared by using a vapor-migration strategy[42] with the Cu loading controlled to be also at ~0.35 wt%, with the single-atom dispersion confirmed by using XAS (Supplementary Fig. 17). Catalytic studies showed that Cu$_1$@C$_3$N$_4$ was merely active for methane oxidation, delivering a

yield of only 0.03% (versus 0.2% by $Cu_2@C_3N_4$) for methyl oxygenates at 50 °C (Fig. 3b). The low reactivity of $Cu_1@C_3N_4$ indicates that Cu monomers, if present in the $Cu_2@C_3N_4$ catalysts, would not make significant contributions to the observed high methane partial oxidation activity, and also underlines the necessity of having dicopper-oxo centers for catalyzing the partial oxidation of methane. The activity of $Cu_2@C_3N_4$ is also substantially enhanced as compared to copper exchanged zeolites. A comparative study of Cu-ZSM5 with a Si/Al ratio of 11.5 and full exchange (Cu/Al ~ 0.51) using similar reaction conditions only delivered a productivity of 25.5 mmol $g_{Cu}^{-1}$ $h^{-1}$ for methyl oxygenates at 50 °C, as compared to 74.4 mmol $g_{Cu}^{-1}$ $h^{-1}$ for $Cu_2@C_3N_4$ at this temperature (Fig. 3b). Note that the copper species in this Cu-ZSM-5 catalyst is also predominantly present in the form of dicopper-oxo centers[36], similar to that identified in $Cu_2@C_3N_4$ (as shown in Fig. 2f). These results thus indicate that the dimeric Cu supported on carbon nitride is much more reactive for the oxidation of methane with $H_2O_2$ than their counterparts confined in zeolites.

In the partial oxidation of methane with $H_2O_2$, the efficiency of utilizing the peroxide oxidizer (instead of producing $O_2$ through a disproportionation reaction) is an important metric for evaluating the performance of catalysts[19,43]. This is usually assessed by comparing the "gain factor" that is defined as the molar ratio between the produced methyl oxygenates ($CH_3OH$ and $CH_3OOH$) and the consumed $H_2O_2$[19]. Post-reaction titration of the concentration of residual hydrogen peroxide using cerium sulfate[44] (Supplementary Fig. 18) showed that the $Cu_2@C_3N_4$ catalyst had a gain factor of 0.19 (Fig. 3c and Supplementary Fig. 19). In comparison, the gain factor was determined to be only 0.03 and 0.06 for $Cu_1@C_3N_4$ and Cu-ZSM5-11.5, respectively. It is interesting that the gain factor exhibited dependence on the copper loading in the dimer catalyst. A $Cu_2@C_3N_4$ catalyst of reduced loading (0.25 wt%) had a gain factor of 0.12, which is lower than that for the normal catalyst with 0.35 wt% of copper. Moreover, correlation between the productivity of methyl oxygenates and the gain factor gives rise to a linear relationship, underscoring its meaning of describing the reactivity between methane and $H_2O_2$ on a given catalyst (Fig. 3c)[19,45].

To understand the enhanced reactivity of $Cu_2@C_3N_4$ for methane partial oxidation, we have performed DFT calculations to simulate the reaction pathways on the dicopper-oxo centers (Figs. 3d, e; also see Supplementary Fig. 20 and Supplementary Table 3). It is predicted that the reaction starts with sequential activation of two $H_2O_2$ molecules on the copper-dimer centers through radical mechanisms[24,46,47]. The first hydrogen peroxide molecule is dissociated via $H_2O_2 \rightarrow \cdot OOH + {}^*H$, where the hydrogen adsorbs on the bridging oxygen and the $\cdot OOH$ radical migrates onto $Cu_\alpha$ to become a peroxyl (${}^*OOH$) adsorbate. The second hydrogen peroxide undergoes $H_2O_2 \rightarrow \cdot OH + {}^*OH$ with the hydroxyl group adsorbing on $Cu_\beta$ and the $\cdot OH$ radical recombines with the ${}^*H$ on the bridging oxygen site to form a $H_2O$ molecule. The involvement of $\cdot OOH$ and $\cdot OH$ in $H_2O_2$ activation was corroborated by the observation of these radicals in the electron paramagnetic resonance (EPR) spectroscopic studies by using 5,5′-Dimethyl-1-pyrroline-N-oxide (DMPO) as the radical trap (Supplementary Fig. 21)[48]. The generation of radicals is the rate limiting factor in both cases of $H_2O_2$ activation, which is predicted to have a kinetic barrier of 0.17 (for $\cdot OOH$) or 0.56 (for $\cdot OH$) eV. Noticeably, these barriers are substantially lower than the corresponding values found for the single-atom Cu sites (1.3 and 1.5 eV, Supplementary Fig. 22) and the dicopper-oxo centers confined in zeolites (0.58 and 0.81 eV in Cu-ZSM-5)[7,49], in line with the higher gain factor and enhanced utilization of $H_2O_2$ as observed on the $Cu_2@C_3N_4$ catalysts (Fig. 3c). The enhanced $H_2O_2$ activation on $Cu_2@C_3N_4$ could be ascribed to the

π-conjugated heterocyclic rings and the semiconducting nature of the $C_3N_4$ substrate, which is known for accommodation of charge transfer and able to supply electrons to the dicopper-oxo center for stabilization of the oxygenated adsorbates[50–53]. The $C_3N_4$ supported Cu dimers are thus believed to be more advantageous than their zeolitic counterparts for catalyzing the redox chemistries being examined here.

Following the activation of $H_2O_2$, methane is introduced to the dicopper-oxo center with one of the C–H bond attacked by the bridging oxygen (Fig. 3d, e). This C–H bond dissociation has a modest energy barrier of 0.61 eV (vs. ~0.71 eV in the case of Cu-ZSM-5)[10,39,40]. While the generated H adsorbs on the bridging oxygen, the methyl group migrates on to the adjacent Cu sites. Hereby the C–H bond dissociation is believed to be *heterolytic* instead of *homolytic* or the *Fenton* type, as no $\cdot CH_3$ radicals were observed using EPR (Supplementary Fig. 23)[19,24,38]. The heterolytic dissociation of C–H bond is believed to be essential for partial oxidation of methane at high selectivities, as the other two activation mechanisms via $\cdot CH_3$ radicals are typically accompanied with over oxidation to form substantial amounts of $CO_2$. Compared to the case in Cu-ZSM-5[37], the Cu dimers supported on g-$C_3N_4$ have shorter Cu-O bond length (1.77 Å vs 1.88 Å) and smaller∠Cu-O-Cu (99.6° vs 135°), which are believed to sterically favor the heterolytic cleavage of the C–H bond and facilitate the transer of the -$CH_3$ group. Noticeably, the -$CH_3$ group can adsorb on either $Cu_\alpha$ or $Cu_\beta$, where the reaction bifurcates into two possible pathways. On the one hand, ${}^*CH_3$ on $Cu_\alpha$ recombine with the ${}^*OOH$ on this site to form ${}^*CH_3OOH$. On the other hand, it can also recombine with ${}^*OH$ on $Cu_\beta$ to form ${}^*CH_3OH$. Desorption of these adsorbates gives rise to the corresponding methyl oxygenates. While the rate is limited by the ${}^*CH_3 + {}^*OH \rightarrow {}^*CH_3OH$ recombination on $Cu_\beta$ (with a barrier of 0.72 eV), the highest barrier for the $CH_3OOH$ pathway is found to be the desorption of ${}^*CH_3OOH$ (0.52 eV). Overall, the $CH_3OOH$ pathway associated with $Cu_\alpha$ is energetically more favorable than the $CH_3OH$ pathway with $Cu_\beta$, explaining the experimentally observed much higher yield of $CH_3OOH$ than $CH_3OH$. The pathways as revealed in Fig. 3d emphasizes the synergy among the two Cu atoms and the bridging O in catalyzing the complex reaction involving multiple molecules (e.g., $CH_4 + 2H_2O_2 \rightarrow CH_3OOH + 2H_2O$), which is a unique feature of the carbon nitride supported dimeric copper centers. An analogous reaction mechanism was also proposed in the partial oxidation of methane with $H_2O_2$ catalyzed by Au-Pd colloids[19].

**Photocatalytic oxidation of methane with $O_2$.** Despite the selective oxidation of methane obtained with $Cu_2@C_3N_4$, the thermocatalytic reaction still relies on the use of $H_2O_2$ as oxidant, which is not a readily available feeding in industry. Moreover, the low $CH_4$ conversions (<1%) also limits the potential of this process for practical implementations. Considering that g-$C_3N_4$ is a semiconductor (with a bandgap of 2.7–2.9 eV[25,26]) with demonstrated photocatalytic applications[27], we have turned to photocatalysis to overcome the limitation of thermocatalytic reactions. Photocatalytic oxidation of methane was carried out at 50 °C by applying near-edge excitation (300 W Xenon lamp equipped with a 420 nm bandpass filter) and using $O_2$ as the oxidant (Supplementary Fig. 24). It is hypothesized that photo-excitation can efficiently activate $O_2$ and generate the oxygenates (${}^*OOH$ and ${}^*OH$), mimicking and improving the role that $H_2O_2$ played in the reaction[21,22,54].

The photocatalytic reaction gave much higher conversions of methane than the thermocatalytic process, reaching 1.3% at 1 h (Fig. 4a). The methane conversion increases with time, reaching

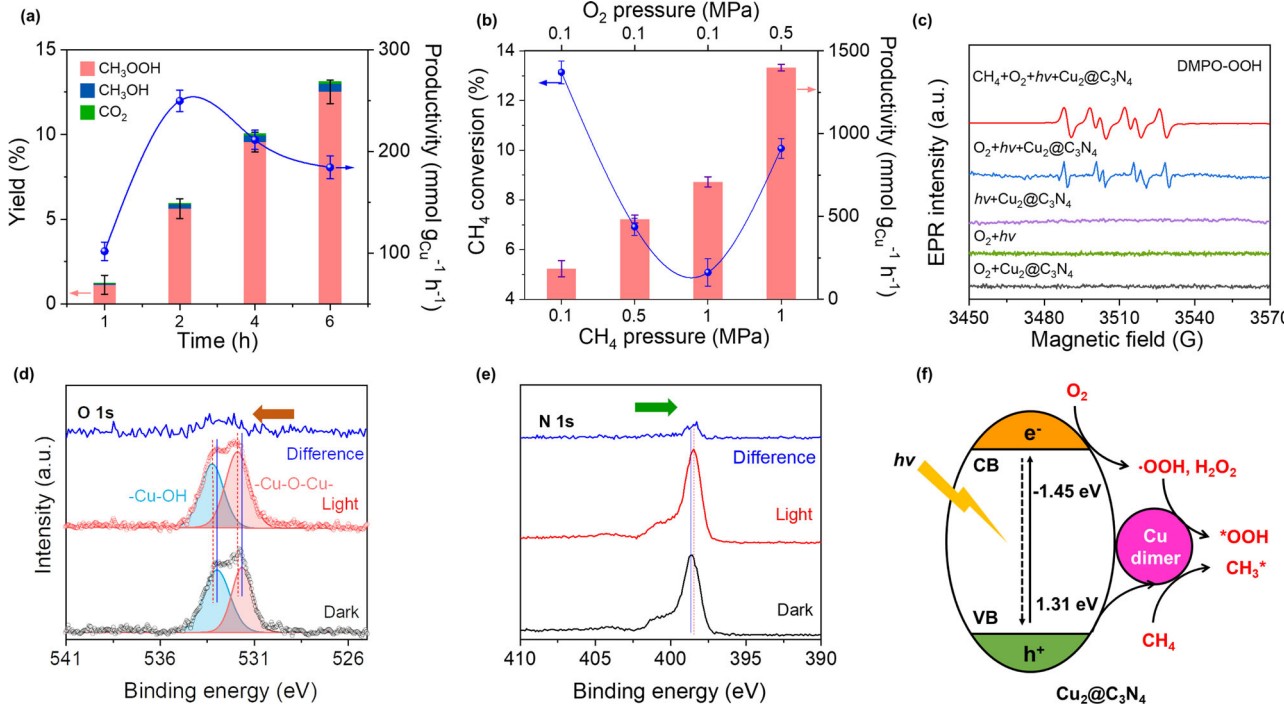

**Fig. 4 Photo-catalytic oxidation of CH₄ with O₂. a** Yields and productivity of methyl oxygenates as a function of reaction time at 0.1 MPa CH₄ and 0.1 MPa O₂. **b** CH₄ conversions and productivity of methyl oxygenates at different CH₄ and O₂ partial pressures. **c** EPR spectra recorded for various control experiments using DMPO as the radical trapping agent. **d, e** In situ irradiation XPS spectra collected at the O 1s (**d**) and N 1s (**e**) edges. **f** Schematic illustration of the photocatalytic oxidation of CH₄ with O₂ catalyzed by Cu₂@C₃N₄. The values "−1.45 and 1.31 eV" label the estimated position of dicopper-oxo states in the band structure of g-C₃N₄, as determined by performing the UV-vis DRS and UPS spectra analysis of Cu₂@C₃N₄. The error bars shown in (**a, b**) indicate the statistical distribution derived from three independent measurements.

~13.1% at 6 h, where the products were found to be still dominated by $CH_3OOH$ and $CH_3OH$ (98.9% selectivity, Supplementary Fig. 25). The productivity of methyl oxygenates reached the peak value of 249.7 mmol $g_{Cu}^{-1}$ $h^{-1}$ at 2 h, representing an improvement factor of ~3.6 as compared to the thermocatalytic reaction. Further improvement of the productivity was obtained by raising the partial pressure of methane ($P_{CH4}$). As $P_{CH4}$ increased from 0.1 to 1 MPa (at $P_{O2}$ = 0.1 MPa, while the total pressure was kept constant at 3 MPa), the productivity escalated from 184.3 to 709.8 mmol $g_{Cu}^{-1}$ $h^{-1}$, albeit with the methane conversion reducing from 13.1 to 5.1 (at 6 h, Fig. 4b and Supplementary Fig. 26). The improvement of productivity at higher $P_{CH4}$ can be ascribed to the increased concentration of dissolved methane in the aqueous solution[20]. The low conversion of methane at high $P_{CH4}$ was likely limited by the inadequacy of oxygen. At $P_{O2}$ = 0.5 MPa and $P_{CH4}$ = 1 MPa, a methane conversion of 10.1% was obtained with >98% selectivity toward methyl oxygenates, corresponding to an even higher productivity of 1399.3 mmol $g_{Cu}^{-1}$ $h^{-1}$. A comprehensive comparison to the literature results under similar reaction conditions indicate that this represents the highest activity for partial oxidation of methane, with improvement factors of at least >10 (Supplementary Table 4).

The photocatalytic oxidation of methane with O₂ was confirmed by conducting control experiments under various conditions (Supplementary Table 5). In particular, the Cu₂@C₃N₄ catalyst was found to be inactive in darkness (while the other conditions were kept the same), ruling out the involvement of thermocatalytic reaction between CH₄ and O₂ in the photocatalytic studies. The photocatalytic activity of bare g-C₃N₄ was also nearly negligible, underlining the role of Cu dimers in catalyzing the related molecular transformations. The generation of active peroxide species in situ during the photocatalytic reaction was confirmed

by performing EPR spectroscopic studies by also using DMPO as the radical trapping agent (Fig. 4c). The spectra recorded under visible light irradiation, in both cases with and without methane, show the fingerprints of ·OOH radicals, which can be assigned to the spin of unpaired electrons on oxygen[21,45,54]. Similar to the findings from photocatalytic studies, such signals were not observed from the controls in the absence of O₂, Cu₂@C₃N₄, or light. These ·OOH radicals are likely derived from the thermal activation of H₂O₂ (as observed in the thermocatalytic studies, Supplementary Fig. 21), which was produced from photocatalytic reduction of O₂ in situ[55–57]. It thus becomes evident that Cu₂@C₃N₄ is not only a good thermocatalyst for partial oxidation of methane with H₂O₂, but also an exceptional photocatalyst when the oxidant is replaced by O₂.

In addition to the reduction of O₂ to peroxides, the photon excitation is also believed to enhance the methane activation. This was revealed by using in situ irradiation X-ray photoelectron spectroscopy (ISI-XPS)[58] to examine charge transfer between the dimeric copper center and the C₃N₄ substrate (see the Experimental Methods in the Supplementary Materials). As shown in Fig. 4d, the XPS spectra collected on hydrated Cu₂@C₃N₄ in darkness exhibited two O 1s peaks at ca. 533.0 and 531.7 eV, which can be assigned to the oxygen binding to Cu, i.e., -Cu-OH and -Cu-O-Cu-, respectively[59,60]. Under light irradiation (400~500 nm), both of these two peaks had a blue shift of ~0.5 eV. Similar observations were obtained at the Cu 2p edge (Supplementary Fig. 27). Meanwhile, a red shift of the N 1s peak associated with C₃N₄ was observed, from 398.8 eV in darkness to 398.2 eV under light irradiation (Fig. 4e). Such phenomena consistently point to the transfer of holes (rather than electrons) from the g-C₃N₄ substrate to the dicopper-oxo center, where CH₄ is activated and oxidized to form *CH₃. Meanwhile, the excited electrons in the g-C₃N₄ substrate lead to the reduction of O₂ and

formation of $H_2O_2$, which then migrates or diffuses onto the dicopper-oxo center and gets activated to form *OOH or *OH. In the following, these oxygen species recombine with *$CH_3$ to form methyl oxygenates, as in the case of thermocatalytic reactions (Fig. 4f and Supplementary Figs. 28–29). Similar phenomena of charge transfer induced catalytic enhancements have previously been reported in photocatalysis using $TiO_2$ based photocatalysts[61–63].

In conclusion, we have developed new dimeric copper catalysts for partial oxidation of methane. These catalysts were synthesized by immobilization of a copper-dimer organometallic complex on graphitic carbon nitride, and dicopper-oxo centers were characterized to anchor on this substrate via Cu-N bonding. The derived $Cu_2@C_3N_4$ catalysts were first examined for thermocatalytic oxidation of methane with $H_2O_2$, and then studied for photocatalytic reactions with $O_2$ being used as the oxidant. Enhanced catalytic activities were demonstrated in both cases as compared to the other reported catalysts under similar reaction conditions, achieving improvement factors of more than an order of magnitude. Synergy of the bridging oxygen, the two copper sites, and the semiconducting $C_3N_4$ substrate has been revealed to promote $H_2O_2$ and $O_2$ activation and the heterolytic scission of $CH_4$. Our work highlights the great potential of carbon nitride supported dimeric copper centers in catalyzing redox chemical reactions.

## Methods

**Materials and Chemicals.** The following chemicals were purchased and used as-received without further purification: Copper(II) chloride dihydrate ($CuCl_2 \cdot 2H_2O$, ACS grade, Sigma Aldrich), 2,2,-bipyridine ($C_{10}H_8N_2$, reagent grade, Sigma Aldrich), oxalic acid ($HO_2CCO_2H$, reagent grade, Alfa Aesar), urea ($NH_2CONH_2$, ACS grade, Sigma Aldrich), dicyandiamide ($NH_2C(=NH)NHCN$, ACS grade, Sigma Aldrich), copper(II) acetylacetonate ($Cu(acac)_2$, ACS reagent, Sigma Aldrich), oleylamine ($CH_3(CH_2)_7CH=CH(CH_2)_8NH_2$, ≥ 98%, Sigma Aldrich), ethanol ($C_2H_5OH$, HPLC grade, Fisher Scientific), methanol ($CH_3OH$, HPLC grade, Fisher Scientific), deionized water (18.2 MΩ) was collected from an ELGA PURELAB flex apparatus.

**Synthesis of copper dimer complex.** Solutions A, B, and C were prepared by ultrasonically dispersion method, respectively. The detailed preparation process was as follows: Solution A: 1.6 mmol 272 mg $CuCl_2 \cdot 2H_2O$ was ultrasonically dispersed in 20 mL deionized water; Solution B: 1.6 mmol 248 mg 2,2,-bipyridine was ultrasonically dispersed in 10 mL methanol; Solution C: 0.8 mmol 100 mg oxalic acid was ultrasonically dispersed in 10 mL deionized water; Subsequently, Adding solution B and solution C to solution A drop by drop respectively and kept stirring for 1 h. Finally, the light-blue solid was obtained by centrifugation, washing with water and methanol for three times and drying in vacuum[29].

**Synthesis of g-$C_3N_4$.** 20 g of urea was placed to an alumina crucible (100 mL). Subsequently, the crucible was sealed with multiple layers of tin foil and put into a muffle furnace with the heating program from 50 °C to 550 °C for 2 h at the rate of 20 °C min$^{-1}$. The obtained powder was further repeated the above calcination operation, the difference was that the heating rate was kept at 5 °C min$^{-1}$ and the retention time at 550 °C was 3 h. Finally, the yellowish-white powder was obtained.

**Synthesis of $Cu_2@C_3N_4$.** Solution A: 0.5 g g-$C_3N_4$ was ultrasonically dispersed in 50 mL methanol solution; Solution B: 42 mg copper dimer was ultrasonically dispersed in 5 mL methanol solution; Solution B was drop-wisely added to solution A and was stirred at room temperature for 24 h, and the obtained solid was calcined in muffle furnace with the heating program from 50 °C to 250 °C for 10 h at the rate of 2 °C min$^{-1}$. Finally, the blue-yellow solid was got.

**Synthesis of $Cu_1@g-C_3N_4$.** [64] 3 g dicyandiamide and 340 mg $CuCl_2 \cdot 2H_2O$ were grounded to be-well mixed, then spread in an alumina crucible (100 mL) with a cap covered. The crucibles were places in a muffle furnace, and gradually heated to 550 °C for 8 h with the ramping rate of 5 °C min$^{-1}$ and then cool down.

**Material characterization.** X-Ray Diffraction (XRD) patterns were obtained from a PANalytical X'Pert3 X-ray diffractometer equipped with a Cu Kα radiation source ($\lambda = 1.5406$ Å). Nitrogen adsorption measurements were measured on a Micromeritics ASAP 2010 instrument with the samples degassed under vacuum at 300 °C for 4 h. Specific surface area (SSA) was calculated using the Brunauer-Emmett-Teller (BET) theory. The Cu contents were determined by inductively coupled plasma mass

spectrometry (ICP-MS) using a PerkinElmer Elan DRC II Quadrupole ICP-MS after dissolution of the samples in aqua regia. High angle annular dark field (HAADF) STEM images were acquired using a JEOL TEM/STEM ARM 200CF (equipped with an Oxford X-max 100TLE windowless X-ray detector) at a 22 mrad probe convergence angle and a 90 mrad inner-detector angle. The analysis of surface elements was performed on X-ray photoelectron spectroscopy (XPS), Thermo Fisher Scientific Escalab 250Xi spectrometer with Al Kα radiation as the excitation source. Fourier Transform Infrared Spectroscopy were carried out on ThermoNicolet Nexus 670. Diffuse reflectance ultraviolet-visible (UV-Vis) spectra were collected on a Shimadzu UV-2450 spectrometer equipped with an integrating sphere attachment using $BaSO_4$ as the reference. FTIR Spectrometer Ultraviolet photoelectron spectroscopy (UPS) measurements were performed on an ESCALAB 250 UPS instrument with a He Iα gas discharge lamp operating at 21.22 eV and a total instrumental energy resolution of 90–120 meV.

XAS experiments were performed at the 10-BM beamline at the Advanced Photon Source (APS) at Argonne National Laboratory. Samples were pressed into a stainless-steel sample holder. All measurements were performed at the Cu K edge (8.9789 keV) in transmission mode in fast scan from 250 eV below the edge to 800 eV above the edge. Spectra processing, including background removal and normalization were performed on ATHENA module in Demeter package. The extraction of structural parameters and fitting of the DFT optimized models of fresh and spent $Cu_2@C_3N_4$ samples were performed on ARTEMIS module. For the optimized structure, EXAFS data were fit from $k = 2.7$ to 10 Å$^{-1}$ (dk = 2) and $R = 1–3.2$ Å with a Hanning window.

Electron Paramagnetic Resonance (EPR) measurements were performed on a Bruker EMX EPR spectrometer at X-band frequency (9.46 GHz). 5,5'-Dimethyl-1-pyrroline-N-oxide (DMPO) was used as the spin-trapping agent, which can capture the radicals •$CH_3$, •OOH and •OH. As for the detection of •OOH and •OH, methanol and DI $H_2O$ were used respectively, due to the DMPO-OOH is not stable in $H_2O$, would be quickly converted to DMPO-OH.

The in-situ irradiation X-Ray photoelectron spectroscopy (ISI-XPS) was carried out on AXIS SUPRA (Kratos Analytical Inc, Shimadazu) coupled with a continuous tunable wavelength light optical fiber (PLS-EM 150, Beijing Perfectlight Co. Ltd.). The wavelength of irradiation light was set at 400–500 nm to mimicking the visible light. The measurement setup is developed to monitor the photoelectron transfer process. Before measurement, the hydrated $Cu_1@g-C_3N_4$ was obtained by pretreatment of fresh $Cu_1@g-C_3N_4$ by water.

## Data availability

The authors declare that the data supporting the findings of this study are available within the paper and its Supplementary Information file. The data generated in this study for main manuscript are provided in the Source Data file. Other raw data of the presented figures and tables are available from the corresponding authors upon request. Source data are provided with this paper.

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

## Acknowledgements

P.X. acknowledges support by the Natural Science Foundation of Zhejiang Province (LR22B060002). C.W. acknowledges support by the Advanced Research Projects Agency – Energy (ARPA-E), Department of Energy (DOE) and the Petroleum Research Fund, American Chemical Society. P.Z. acknowledges support by National Natural Science Foundation of China (51972287, U2004172). J.D., Z.Y. and S.D. acknowledge support by The Funds for National Natural Science Foundation of China (21706131, 22008211, and 22178309). R. S.-Y. acknowledges financial support from National Science Foundation award DMR-1809439.

## Author contributions

C.W. and P.X. contributed to the idea and experimental design. P.X., J.D., and J.Z. conducted the synthesis of control samples, catalytic evaluation. T. Pu performed spectroscopic analysis. N.Z-F, H.Z., and S.D. conducted the EPR characterization and NMR analysis. C.W., Z.H. and R. S-Y. performed the high resolution microscopy. Z.Y. and S.D. contributed to the simulation analysis. P.Z. conducted the in-situ irradiation X-Ray photoelectron spectroscopy characterization. C.W. and P.X. wrote the paper and all authors commented on the final manuscript.

## Competing interests

The authors declare no competing interests.
