## [Peer review file · Nature Communications]

REVIEWER COMMENTS

Reviewer #1 (Remarks to the Author):

The authors have reported a new method to successfully synthesize oxo dicopper ($[\text{Cu-O-Cu}]^{2+}$) anchored on graphitic carbon nitride. The synthesized $\text{Cu}_2@\text{C}_3\text{N}_4$ catalyst exhibits high selectivity for partial oxidation of methane under thermo- and photo-catalytic conditions. Many characterizations have been conducted to assure the structures of $\text{Cu}_2@\text{C}_3\text{N}_4$ catalyst. It is an interesting work, I recommend the paper for publication after addressing the following issues.

1. There are lots of single atoms in the STEM images. Therefore, single atom Cu in C_3N_4 may affect the bond number analysis of EXAFS. How to exclude the single atoms contribution in the EXAFS characterizations.
2. How to exclude the single atoms contribution to the activity of partial oxidation of methane?
3. Two hours catalyst test is not stable. Is the dicopper catalyst stable during the partial oxidation of methane reaction for a long time?
4. How to calculate the formation energies in Table S1? More calculation details have to be given properly.
5. More DFT calculations, such as XPS and FTIR calculations, have to be performed to compare with experiment measurements to affirm the $\text{Cu}_2@\text{C}_3\text{N}_4$ structure.
6. The meaning of energetic values in the reaction cycle in Figure 3 have to be illustrated clearly. It is better to present a potential energy surface diagram in the manuscript.

Reviewer #2 (Remarks to the Author):

This is an interesting paper. The topic of selective methane oxidation continues to attract a great deal of attention. This paper concerns a copper catalyst that uses hydrogen peroxide as the oxidant for thermal oxidation. This makes up the bulk of the paper. There is a short section on using oxygen with photocatalysis.

Cu is well known as an active site for methane oxidation. The novelty here is the use of a specially prepared Cu dimer catalyst. There is clear evidence that this is present in the fresh catalyst but there is no evidence presented as to whether this structure persists during or after the reaction. Does the complex break down and leach Cu into solution which would be a very good catalyst for making the reactive oxygen species OOH and OH radicals that they cite as the oxidising species. So they will need to provide evidence that the catalyst structure remains intact during the reaction and that it is the sole species responsible for the observed activity. By the way the mechanism was originally proposed in reference 19 and that should be acknowledged.

The stability of hydrogen peroxide at 30 C and 70 C is questionable in the reaction times stated.

Indeed, the "gain factor" is really low so what is the oxidising species. If it is OH or OOH these radicals are very short lived so there are questions remaining on this aspect.

Reviewer #3 (Remarks to the Author):

This work reports on the preparation of dicopper oxocenters anchored on carbon nitride and its utilization in the selective oxidation of methane to oxygenated products, through thermo- and photo-catalytic approaches. The results indicate the formation of Cu dimers anchored on nitrogen groups, with particular structural properties and intermediate oxidation states. These materials demonstrate to reach relatively high methane conversion values under the evaluated conditions, by retaining high selectivity towards CH_3OOH and CH_3OH , over CO_2 , especially in the photocatalytic tests. Moreover, the authors properly correlate the intrinsic catalytic activation of H_2O_2 or O_2 , in the thermo- and photo-catalytic approaches, respectively.

Certainly, the results are sound, and significant effort is done, in order to elucidate the correlation between activity and structure. However, some points indicated below could serve to clarify or

improve the manuscript:

-First of all, the Introduction section can be improved in order to justify the selection of materials (Cu dimers and C₃N₄), in light of the hypothesis posed in page 9. "We hypothesize that their non-integer oxidation state (intermediate between +1 and +2) and reduced cluster size (smaller Cu-Cu distance as compared to Cu-ZSM-5) would lend them exquisite catalytic performance for selective oxidation of methane".

-Concerning the comparison done in the thermo-catalytic approach with Cu-ZSM-5, although both have the dicopper-oxo centers, Cu oxidation states are different. Is there any other particular difference, such as Cu content, that could potentially have influence on the comparative results?

-In the thermocatalytic approach, the authors demonstrate that gain factor, productivity and Cu content (of Cu oxocenters) are correlated, being sample with 0.35% the one with the highest oxygenated productivity. Has any additional attempt or preliminary test been carried out to increase Cu content and determine its effect on productivity?

-The proposed photocatalytic mechanism (Figure 4f) is not clear. Electron transfer from C₃N₄ to Cu dimer is suggested to take place, based on in situ XPS results, while holes are also illustrated to be transferred to the dimer, to oxidize CH₄ leading to the formation of *CH₃. Although all such processes would have specific dynamics, that mechanism would most likely lead to hole-electron recombination (comparable to type I heterojunctions). Please check, comment or clarify. Also, what do the values "-1.45 and 1.31 eV" represent?

Other minor comments:

-Please indicate in Figure S1 what a) and b) parts correspond to.

-At what temperature was the longer experiment in Figure S13b carried out?

-In last paragraph of page 11, please change Figure S2 by Figure S20.

-Please check: conversions are not shown in Figure 4a, but Yield (%), and the values at 1h are not included, nor selectivity.

Point-by-Point Response to Reviewers' Comments

Reviewer #1:

The authors have reported a new method to successfully synthesize oxo dicopper ($[\text{Cu-O-Cu}]^{2+}$) anchored on graphitic carbon nitride. The synthesized $\text{Cu}_2@C_3N_4$ catalyst exhibits high selectivity for partial oxidation of methane under thermos- and photo-catalytic conditions. Many characterizations have been conducted to assure the structures of $\text{Cu}_2@C_3N_4$ catalyst. It is an interesting work, I recommend the paper for publication after addressing the following issues.

Response: We thank the reviewer for recommending publication of our work. We have followed the reviewer's suggestions to revise the manuscript and addressed the comments as detailed below.

Q1. There are lots of single atoms in the STEM images. Therefore, single atom Cu in C_3N_4 may affect the bond number analysis of EXAFS. How to exclude the single atoms contribution in the EXAFS characterizations.

Response: We agree with the reviewer that we cannot fully exclude the presence of single-atom Cu in the $\text{Cu}_2@C_3N_4$ catalysts but believe Cu dimers predominate in our case. Figure R1 below shows the comparison of EXAFS profiles for $\text{Cu}_1@C_3N_4$ and $\text{Cu}_2@C_3N_4$ (presented as a part of Figure S17 in the revised manuscript). Compared to $\text{Cu}_2@C_3N_4$, $\text{Cu}_1@C_3N_4$ lacks the second-shell Cu-Cu scattering feature at $\sim 2.3 \text{ \AA}$ (prior to phase correction), while the primary Cu-N bonding (below 2 \AA) feature is quite similar for the two catalysts. Motivated by the reviewer's comment, we have applied a linear combination of Cu monomers and dimers to fit the EXAFS spectrum of $\text{Cu}_2@C_3N_4$ and derived a ratio of 72.4% for the Cu atoms in the dimeric configuration, versus 27.6% for monomers (Table 1, Figure 2d and Figure S6). This finding is largely in line with the statistical analysis based on STEM images, where the monomers and dimers were estimated to occupy ca. 46% and 54%, respectively, of the counts (corresponding to $\sim 30\%$ of Cu atoms in monomeric configuration). In the revised manuscript and supporting information, we have revised Table 1, Figure 2d, Figure S6 and the corresponding discussions accordingly.

Figure R1. The Comparison of k^2 -weighted EXAFS spectra at Cu K edge for $\text{Cu}_1@C_3N_4$ and $\text{Cu}_2@C_3N_4$.

Table 1. Structural parameters being used to fit the EXAFS spectrum for $\text{Cu}_2@\text{C}_3\text{N}_4$ with consideration of both monomeric and dimeric Cu sites.

Scattering path	CN	Distance (Å)	$\sigma^2(\text{Å}^2)$	R-factor
Cu-O	1.25 ± 0.20	1.77 ± 0.01		
Cu-N (dimer)	1.84 ± 0.29	1.99 ± 0.01	0.0056 ± 0.0006	0.009 ± 0.001
Cu-N (single atom)	2.42 ± 0.32	1.98 ± 0.01		
Cu-Cu	0.88 ± 0.15	2.71 ± 0.01	0.0038 ± 0.0004	

Figure 2d. Fitting of the EXAFS spectrum for $\text{Cu}_2@\text{C}_3\text{N}_4$ with consideration of both monomeric and dimeric Cu sites.

Q2. How to exclude the single atoms contribution to the activity of partial oxidation of methane?

Response: Considering that monomeric Cu in copper exchanged zeolites or supported on other substrates has also been reported to be active for CH_4 oxidation,¹⁻³ we have conducted control experiments on single-atom $\text{Cu}_1@\text{C}_3\text{N}_4$ (Figure S17) for the catalytic studies. It was found that $\text{Cu}_1@\text{C}_3\text{N}_4$ was merely active for methane oxidation, e.g., delivering a yield of only 0.03% (versus 0.2% by $\text{Cu}_2@\text{C}_3\text{N}_4$) for methyl oxygenates at 50 °C (Figure 3b). The low reactivity of Cu monomers in partial oxidation of methane is further supported by the much lower gain factor of $\text{Cu}_1@\text{C}_3\text{N}_4$ (0.03) than $\text{Cu}_2@\text{C}_3\text{N}_4$ (0.19) for H_2O_2 utilization in partial oxidation of methane, which indicates H_2O_2 activation is more challenging on Cu monomers (Figure 3c and Figure S19). These results suggest that Cu monomers, if present in the $\text{Cu}_2@\text{C}_3\text{N}_4$ catalysts, would not make significant contributions to the observed high methane partial oxidation activity, and also underlines the necessity of having dicopper-oxo centers for catalyzing the partial oxidation of methane under the reaction conditions investigated in this work.

Q3. Two hours catalyst test is not stable. Is the dicopper catalyst stable during the partial oxidation of methane reaction for a long time?

Response: Motivated by the reviewer's comment, we have run durability tests with elongated periods. The methane conversions and product selectivities were implemented to Figure S13b of the revised manuscript. The methane conversion was found to increase monotonically with the reaction time. Although more CO₂ was obtained at elongated reaction times, the selectivity toward methyl oxygenates was still as high as >82% after 6 h, indicating that the Cu₂@C₃N₄ catalysts are stable under the reaction conditions (at 50 °C). The increase yield of CO₂ can be attributed to the facile over-oxidation of methyl oxygenates as thermodynamically CO₂ is the most stable product. Similar phenomena were also reported in the literature.^{4, 5} Notably, ICP analysis of the reaction solution after the 6-h durability test showed no Cu signal, ruling out the possibility of metal leaching during the reaction.

Moreover, we have characterized the Cu₂@C₃N₄ catalyst after the 6-h durability test by using aberration-corrected high-angle annular darkfield scanning transmission electron microscopy (HAADF-STEM) and X-ray absorption (XAS) spectroscopy. The collected STEM images still exhibited a lot of adjacent, paired bright dots that can be assigned to Cu dimers (Figure S15a). EXAFS analysis for this spent Cu₂@C₃N₄ catalyst, again with consideration of both monomeric and dimeric Cu sites, showed the coordination configuration was largely preserved (Table S2, in comparison to Table 1). These results thus confirm the high stability of Cu₂@C₃N₄ under the methane partial oxidation reaction conditions.

Figure S13b. Time-dependent CH₄ conversions (blue dots) and product selectivities (colored histograms) for the thermo-catalytic oxidation of CH₄ with H₂O₂ at 50 °C.

Table S2. Structural parameters being used to fit the EXAFS spectrum for spent Cu₂@C₃N₄ after reaction with considering both of the contribution of single and dimeric Cu sites.

Scattering path	CN	Distance (Å)	σ ² (Å ²)	R-factor
-----------------	----	--------------	----------------------------------	----------

Cu-O	1.35±0.24	1.77±0.01		
Cu-N (dimer)	1.91±0.33	1.99±0.01	0.0049±0.0005	0.008±0.001
Cu-N (single atom)	2.42±0.32	1.98±0.01		
Cu-Cu	0.81±0.14	2.71±0.01	0.0043±0.0004	

Figure S15. Characterizations of the spent $\text{Cu}_2@C_3N_4$ catalyst after the 6-h durability test. (a) Representative HAADF-STEM images with Cu dimers highlighted using red circles. (b) k^2 -weighted EXAFS spectra at the Cu K edge, with Cu foil, Cu_2O , CuO and Cu-TPP (one Cu coordinated with for N atoms) being used as the references. (c, d) Fitting of the EXAFS spectrum with consideration of both monomeric and dimeric Cu sites.

Q4. How to calculate the formation energies in Table S1? More calculation details have to be given properly.

Response: To calculate the formation energies of potential Cu dimer structures, we used the equation

$$E(\text{formation energy}) = E(\text{System}) - E(\text{Substrate}) - a \times E(\text{Cu}) - b \times 0.5 \times E(\text{O}_2)$$

Here, $E(\text{System})$ represents the total free energy of the system, $E(\text{Substrate})$, $E(\text{Cu})$ and $E(\text{O}_2)$ represent the free energies of the support, Cu atoms and oxygen gas, and a and b are the numbers of Cu and O atoms involved in the considered structure. We have added such information to Table S1 in the revised manuscript.

Q5. More DFT calculations, such as XPS and FTIR calculations, have to be performed to compare with experiment measurements to affirm the $\text{Cu}_2@\text{C}_3\text{N}_4$ structure.

Response: Following the review's suggestion, we calculated the Bader charge of Cu in $\text{Cu}_2@\text{C}_3\text{N}_4$. It was found that the Cu atoms have oxidation states of +1.63 and +1.72 (Figure S7), resembling the results derived from XPS and XANES spectra with an intermediate oxidation state between +1 and +2.

The Cu dimers supported on C_3N_4 are hard to be detected by using FTIR, due to low loading of Cu (0.35 wt%) and the strong absorption of the C_3N_4 substrate. Instead, we have compared the experimental and calculated FTIR spectra for $[\text{Cu}_2(\text{bpy})_2(\mu\text{-ox})]\text{Cl}_2$, 2,2'-bipyridine and oxalic acid, in order to confirm the molecular transformation in the synthesis of $\text{Cu}_2@\text{C}_3\text{N}_4$. The calculation was performed using the basis set of B3LYP/6-31g with Gaussian broadening. The simulated IR spectra (Figure R2) are consistent with the experimental results (Figure 1b). For instance, the asymmetric stretching vibration of C=C in the pyridyl ring of $[\text{Cu}_2(\text{bpy})_2(\mu\text{-ox})]\text{Cl}_2$ was predicted to give a FTIR peak at $\sim 1630\text{ cm}^{-1}$, close to the experimental result at 1650 cm^{-1} .

Figure R2. The calculated IR Spectra of copper-dimer precursor $[\text{Cu}_2(\text{bpy})_2(\mu\text{-ox})]\text{Cl}_2$, 2,2'-bipyridine and oxalic acid by gaussian simulation.

Q6. The meaning of energetic values in the reaction cycle in Figure 3 have to be illustrated clearly. It is better to present a potential energy surface diagram in the manuscript.

Response: We thank the reviewer for pointing out this issue. The free energy diagram shown below was implemented as Figure 3e in the revised manuscript to illustrate the reaction pathway,

with the energies given in Figure 3d representing the energy barriers associated with the given molecular transformations. For example, dissociations of the first (I-1 to I-2) and second (I-2 to I-3) H_2O_2 molecule has an energy barrier of 0.17 and 0.56 eV, respectively, corresponding to the transition states TS1 and TS2 in the predicted reaction pathway.

Figure 3e. Predicted free energy diagram for the partial oxidation of CH_4 with H_2O_2 on $\text{Cu}_2@\text{C}_3\text{N}_4$. Three stages consisting of H_2O_2 activation, CH_4 activation and methyl oxygenates formation are marked in different colors.

Reviewer #2:

This is an interesting paper. The topic of selective methane oxidation continues to attract a great deal of attention. This paper concerns a copper catalyst that uses hydrogen peroxide as the oxidant for thermal oxidation. This makes up the bulk of the paper. There is a short section on using oxygen with photocatalysis.

Response: We thank the reviewer for his/her interest in our work. We have followed the reviewer's suggestions to revise the manuscript and addressed the comments as detailed below.

Q1. Cu is well known as an active site for methane oxidation. The novelty here is the use of a specially prepared Cu dimer catalyst. There is clear evidence that this is present in the fresh catalyst but there is no evidence presented as to whether this structure persists during or after the reaction. Does the complex break down and leach Cu into solution which would be a very good catalyst for making the reactive oxygen species OOH and OH radicals that they cite as the oxidising species. So they will need to provide evidence that the catalyst structure remains intact during the reaction and that it is the sole species responsible for the observed activity. By the way the mechanism was originally proposed in reference 19 and that should be acknowledged.

Response: Motivated by the reviewer's comment, we have performed ICP measurements on the spent $\text{Cu}_2@\text{C}_3\text{N}_4$ catalyst (after 6 h of reaction, Figure S13b) and the solution after the reaction and confirmed no Cu leaching was detected. We have also characterized the spent catalysts using STEM and XAS. Similar to the situation prior to the catalytic studies, STEM images of the spent catalysts also exhibit paired bright dots, indicating the preservation of Cu dimers after the reaction (Figure S15a in the revised manuscript). EXAFS analysis also showed that the atomic structure (including coordination number and bonding length) of Cu species in the spent catalysts (Figure S15b-S15d and Table S2) is consistent with the fresh ones (Figure 2 and Table 1). These results thus confirmed the stability of $\text{Cu}_2@\text{C}_3\text{N}_4$ under the reaction conditions, with the dioxo-copper centers being the active sites for partial oxidation of methane. In the revised manuscript, we have cited ref. 19 and acknowledged the originally proposed mechanism on page 13.

Figure S13b. Time-dependent CH_4 conversions (blue dots) and product selectivities (colored histograms) for the thermo-catalytic oxidation of CH_4 with H_2O_2 at 50°C .

Table S2. Structural parameters being used to fit the EXAFS spectrum for spent $\text{Cu}_2@\text{C}_3\text{N}_4$ after reaction with considering both of the contribution of single and dimeric Cu sites.

Scattering path	CN	Distance (\AA)	$\sigma^2(\text{\AA}^2)$	R-factor
Cu-O	1.35 ± 0.24	1.77 ± 0.01		
Cu-N (dimer)	1.91 ± 0.33	1.99 ± 0.01	0.0049 ± 0.0005	0.008 ± 0.001
Cu-N (single atom)	2.42 ± 0.32	1.98 ± 0.01		
Cu-Cu	0.81 ± 0.14	2.71 ± 0.01	0.0043 ± 0.0004	

Figure S15. Characterizations of the spent $\text{Cu}_2@\text{C}_3\text{N}_4$ catalyst after the 6-h durability test. (a) Representative HAADF-STEM images with Cu dimers highlighted using red circles. (b) k^2 -weighted EXAFS spectra at the Cu K edge, with Cu foil, Cu_2O , CuO and Cu-TPP (one Cu coordinated with for N atoms) being used as the references. (c, d) Fitting of the EXAFS spectrum with consideration of both monomeric and dimeric Cu sites.

Q2. The stability of hydrogen peroxide at 30 C and 70 C is questionable in the reaction times stated. Indeed, the "gain factor" is really low so what is the oxidising species. If it is OH or OOH these radicals are very short lived so there are questions remaining on this aspect.

Response: Although thermodynamically H_2O_2 is unstable and would spontaneously decompose into H_2O and O_2 at 30 or 70 °C (with ΔG of -152.5 and -153.7 kJ mol^{-1} , respectively), it is kinetically stable at these temperatures due to the relatively large dissociation barriers of ~0.6 eV on Cu dimers or ~1.2 eV on Cu monomers. The post-reaction titration of residual hydrogen peroxide using cerium sulfate showed that the conversion of H_2O_2 was as low as 6.2% at 30 °C and 12.4% at 70 °C, confirming its relatively high stability under the methane partial oxidation reaction conditions.

We have performed additional control experiments at 50 °C using H_2O or O_2 as the oxidizing agent (Figure R3). No oxidation products were detected in these cases, confirming H_2O_2 as the oxidizer for the partial oxidation of methane. We would like to point out that $\cdot\text{OOH}$ and $\cdot\text{OH}$ radicals are possible reaction intermediates derived from the dissociative activation of H_2O_2 on the $\text{Cu}_2@\text{C}_3\text{N}_4$ catalysts, as detected using electron paramagnetic resonance (EPR) spectroscopy in the control experiments (with DMPO being used as the radical trapping agent, Figure S21) and confirmed by the reaction pathway predicted based on DFT calculations (Figure 3). However, they are not necessarily the active species to attack CH_4 , which would otherwise cause kinetic hindrance due to the short lives of these radical species, as pointed out by the reviewer. Instead, in the predicted reaction pathway (Figure 3e), the Cu-dimer catalysts activate H_2O_2 and CH_4 sequentially to form $\cdot\text{OH}$ (via $\cdot\text{OH}$), $\cdot\text{OOH}$ (via $\cdot\text{OOH}$) and $\cdot\text{CH}_3$, which then recombine to form methyl oxygenates. This pathway is not subjected to the possible kinetic limitation associated with the short-lived radicals.

Figure R3. The comparison of catalytic activity for CH_4 partial oxidation by using different oxidizing agents.

Reviewer #3 (Remarks to the Author):

This work reports on the preparation of dicopper oxocenters anchored on carbon nitride and its utilization in the selective oxidation of methane to oxygenated products, through thermo- and photo-catalytic approaches. The results indicate the formation of Cu dimers anchored on nitrogen groups, with particular structural properties and intermediate oxidation states. These materials demonstrate to reach relatively high methane conversion values under the evaluated conditions, by retaining high selectivity towards CH₃OOH and CH₃OH, over CO₂, especially in the photocatalytic tests. Moreover, the authors properly correlate the intrinsic catalytic activation of H₂O₂ or O₂, in the thermo- and photo-catalytic approaches, respectively.

Certainly, the results are sound, and significant effort is done, in order to elucidate the correlation between activity and structure. However, some points indicated below could serve to clarify or improve the manuscript:

Response: We thank the reviewer for recognizing our work. We have addressed the reviewer's comments in the revised manuscript, with the details provided below.

Q1. -First of all, the Introduction section can be improved in order to justify the selection of materials (Cu dimers and C₃N₄), in light of the hypothesis posed in page 9. "We hypothesize that their non-integer oxidation state (intermediate between +1 and +2) and reduced cluster size (smaller Cu-Cu distance as compared to Cu-ZSM-5) would lend them exquisite catalytic performance for selective oxidation of methane".

Response: We thank the reviewer for this constructive suggestion. We have revised the second paragraph of Introduction to rationalize the selection of C₃N₄ as substrate to support Cu dimers: *"In this aspect, graphitic carbon nitride (g-C₃N₄) represents a promising photocatalytic substrate with a modest band gap in the range of 2.7-2.9 eV.⁶⁻⁸ Its abundant nitrogen sites have been shown in many reports to be capable of anchor atomically dispersed transition metal sites.⁹ It thus becomes interesting to investigate the potential coordination of active Cu sites on g-C₃N₄ and examine their synergies in partial oxidation of methane."*

Q2. Concerning the comparison done in the thermo-catalytic approach with Cu-ZSM-5, although both have the dicopper-oxo centers, Cu oxidation states are different. Is there any other particular difference, such as Cu content, that could potentially have influence on the comparative results?

Response: We thank the reviewer for pointing out this question and agree that the dicopper-oxo centers in Cu₂@C₃N₄ have multiple distinct atomic structures and electronic chemical properties from the counterpart in Cu-ZSM-5.

Besides the Cu sites exhibit different oxidation states (+1.7 vs +2), our combined EXAFS analysis and DFT calculations resolved the Cu-Cu distance to be ~2.71 Å (Table 1). It is noted that this value is much smaller than the Cu-Cu distance (~4.10 Å) associated with the dicopper-oxo center in Cu-ZSM-5.¹⁰ In addition, compared to the case in Cu-ZSM-5, the Cu dimers supported on g-C₃N₄ have shorter Cu-O bond length (1.77 Å vs 1.88 Å) and smaller ∠ Cu-O-Cu (99.6° vs 135°). The distinct structure of Cu dimer in Cu₂@C₃N₄ is believed to sterically favor the heterolytic cleavage of the C-H bond and facilitate the transfer of the -CH₃ group for CH₄ activation, evidenced by a lower energy barrier for the C-H bond dissociation (0.61 eV on Cu₂@C₃N₄) than for Cu-ZSM-5 (~0.71 eV).^{11, 12}

Moreover, the advantage of Cu₂@C₃N₄ (vs Cu-ZSM-5) could also be attributed to the π-conjugated heterocyclic rings and the semiconducting nature of the g-C₃N₄ substrate, which is

known for accommodation of charge transfer between it and the anchored transition metal sites, which is beneficial for stabilization of the oxygenated adsorbates involved in the partial oxidation of methane with H₂O₂. Our DFT calculations show that the energy barriers for H₂O₂ activation to form ·OOH and ·OH are only 0.17 and 0.56 eV, respectively, substantially lower than the corresponding values (0.58 and 0.81 eV for ·OOH and ·OH, respectively)^{4, 13} found for the dicopper-oxo centers confined in Cu-ZSM-5.

Q3. -In the thermocatalytic approach, the authors demonstrate that gain factor, productivity and Cu content (of Cu oxocenters) are correlated, being sample with 0.35% the one with the highest oxygenated productivity. Has any additional attempt or preliminary test been carried out to increase Cu content and determine its effect on productivity?

Response: We indeed synthesized three Cu₂@C₃N₄ catalysts with 0.25, 0.35 and 0.90 wt% of Cu to study the Cu loading effect on the productivity of methyl oxygenates (Figure R4). The reactivity maximizes at 0.35 wt% Cu. Although we still cannot exclude the possibility of achieving higher catalytic activity at a higher (but lower than 0.90 wt%) loading, we believe adding more Cu species may cause cauterization of Cu and reduction of productivity.¹⁴

Figure R4. The productivity of methyl oxygenates depended on the Cu loadings.

Q4. -The proposed photocatalytic mechanism (Figure 4f) is not clear. Electron transfer from C₃N₄ to Cu dimer is suggested to take place, based on *in situ* XPS results, while holes are also illustrated to be transferred to the dimer, to oxidize CH₄ leading to the formation of *CH₃. Although all such processes would have specific dynamics, that mechanism would most likely lead to hole-electron recombination (comparable to type I heterojunctions). Please check, comment or clarify. Also, what do the values "-1.45 and 1.31 eV" represent?

Response: We thank the reviewer for raising this question. Charge transfer between the dimeric copper center and the C₃N₄ substrate was revealed by using *in situ* irradiation X-ray photoelectron spectroscopy (ISI-XPS). As shown in Figure 4d and Figure S27, shifts toward higher energies were observed at both O 1s and Cu 2p edges upon light irradiation, which was accompanied with a shift of the N 1s peak toward lower binding energies. Such phenomena consistently point to the transfer of holes (rather than electrons) from the g-C₃N₄ substrate to the dicopper-oxo center,

where CH_4 is activated and oxidized to form $^*\text{CH}_3$. Meanwhile, the excited electrons in the g- C_3N_4 substrate lead to the reduction of O_2 to H_2O_2 , which then migrates or diffuses onto the dicopper-oxo center and gets activated to form $^*\text{OOH}$ or $^*\text{OH}$. In the following, these oxygen species recombine with $^*\text{CH}_3$ to form methyl oxygenates, as in the case of thermocatalytic reactions. We have added the corresponding discussion to the revised manuscript (page 16).

The values “-1.45 and 1.31 eV” label the estimated position of dicopper-oxo states in the band structure of g- C_3N_4 . This was determined by performing Tauc plot analysis on the UV-vis DRS and UPS spectra,^{15, 16} as shown in Figures S28 and S29. In the revised manuscript, we have revised the caption of Figure 4f to specify this.

Other minor comments:

Q5. -Please indicate in Figure S1 what a) and b) parts correspond to.

Response: Figure S1 exhibits the weight loss and the corresponding first derivative values of the dimeric copper complex (a) or the pristine $\text{Cu}_2@ \text{C}_3\text{N}_4$ with ligands (b) when burned in air during thermogravimetric analysis (TGA). We have revised the corresponding figure caption accordingly.

Q6. -At what temperature was the longer experiment in Figure S13b carried out?

Response: The longer experiment was carried out at 50 °C. We have added the temperature info to the caption of Figure S13 in the revised manuscript.

Q7. -In last paragraph of page 11, please change Figure S2 by Figure S20.

Response: We thank the reviewer for pointing out this error. We have corrected the corresponding figure citation.

Q8. -Please check: conversions are not shown in Figure 4a, but Yield (%), and the values at 1h are not included, nor selectivity.

Response: We added the yield (at 1 h) in Figure 4a, and also put the CH_4 conversions and selectivity of methyl oxygenates depending on time in the Supporting Information as Figure S25.

Reference

1. Wu, B. et al. Cu single-atoms embedded in porous carbon nitride for selective oxidation of methane to oxygenates. *Chem. Commun.* **56**, 14677-14680 (2020).
2. Grundner, S., Luo, W., Sanchez-Sanchez, M. & Lercher, J.A. Synthesis of single-site copper catalysts for methane partial oxidation. *Chem. Commun.* **52**, 2553-2556 (2016).
3. Tang, X. et al. Direct oxidation of methane to oxygenates on supported single Cu atom catalyst. *Appl. Catal. B* **285** (2021).
4. Hammond, C. et al. Direct Catalytic Conversion of Methane to Methanol in an Aqueous Medium by using Copper-Promoted Fe-ZSM-5. *Angew. Chem. Int. Edit.* **51**, 5129-5133 (2012).
5. Agarwal, N.F., S. J. McVicker, R. U. Althahban, S. M. Dimitratos, N. He, Q. Morgan, D. J. Jenkins, R. L. Willock, D. J. Taylor, S. H. Kiely, C. J. Hutchings, G. J. Aqueous Au-Pd colloids catalyze selective CH₄ oxidation to CH₃OH with O₂ under mild conditions. *Science* **358**, 223-226 (2017).
6. Wen, J.Q.X., J. Chen, X. B. Li, X. A review on g-C₃N₄-based photocatalysts. *Appl. Surf. Sci.* **391**, 72-123 (2017).
7. Xu, Y. & Gao, S.P. Band gap of C₃N₄ in the GW approximation. *Int. J. Hydrogen Energ.* **37**, 11072-11080 (2012).
8. Su, F.Z. et al. mpg-C₃N₄-Catalyzed Selective Oxidation of Alcohols Using O₂ and Visible Light. *J. Am. Chem. Soc.* **132**, 16299-16301 (2010).
9. Shi, Z.S., Yang, W.Q., Gu, Y.T., Liao, T. & Sun, Z.Q. Metal-Nitrogen-Doped Carbon Materials as Highly Efficient Catalysts: Progress and Rational Design. *Adv. Sci.* **7** (2020).
10. Tsai, M.L. et al. [Cu₂O]²⁺ Active Site Formation in Cu-ZSM-5: Geometric and Electronic Structure Requirements for N₂O Activation. *J. Am. Chem. Soc.* **136**, 3522-3529 (2014).
11. Woertink, J.S. et al. A [Cu₂O]²⁺ core in Cu-ZSM-5, the active site in the oxidation of methane to methanol. *P. Natl. Acad. Sci. USA* **106**, 18908-18913 (2009).
12. Kulkarni, A.R., Zhao, Z.J., Siahrostami, S., Norskov, J.K. & Studt, F. Cation-exchanged zeolites for the selective oxidation of methane to methanol. *Catal. Sci. Technol.* **8**, 114-123 (2018).
13. Hori, Y., Shiota, Y., Tsuji, T., Kodera, M. & Yoshizawa, K. Catalytic Performance of a Dicopper-Oxo Complex for Methane Hydroxylation. *Inorg. Chem.* **57**, 8-11 (2018).
14. Xie, P.F. et al. Bridging adsorption analytics and catalytic kinetics for metal-exchanged zeolites. *Nat. Catal.* **4**, 144-156 (2021).
15. Luo, L.H. et al. Water enables mild oxidation of methane to methanol on gold single-atom catalysts. *Nat. Commun.* **12**, 1218 (2021).
16. Liu, J. et al. Metal-free efficient photocatalyst for stable visible water splitting via a two-electron pathway. *Science* **347**, 970-974 (2015).

REVIEWERS' COMMENTS

Reviewer #1 (Remarks to the Author):

The authors have revised the paper accordingly and the paper can be publishable.

Reviewer #2 (Remarks to the Author):

I am largely satisfied with the responses and revisions made. The paper is now publishable. I disagree that hydrogen peroxide is kinetically stable at 50 or 70C so this part needs more thought by the authors

Reviewer #3 (Remarks to the Author):

As stated in the review of previous version, this is a sound and interesting work dealing with the preparation of Cu-dimers anchored on carbon nitride structure, with significant activity towards methane oxidation to oxygenated products, through thermal and photocatalytic routes. In this revised version, the authors have addressed individual concerns and points related to the catalytic role, stability of dicopper centers and reaction mechanism. In this regard, the authors have conducted longer experiments proving the relative stability of these structures, and have provided more evidence of the proposed mechanism and intrinsic activities, including theoretical calculations. Also, a better explanation of the photocatalytic process has been added, with some experimental insights on the charge transfer from the semiconductor to the catalytic sites, assisted by thermal conditions.

In general, this is a complete work with conclusive evidences of the favored role of catalytic centers towards scission of stable CH₄ molecule and additional steps promoting the activation of the oxidizing species. Therefore, it not only highlights the intrinsic activity of the proposed structures, but also provides ideas for development of alternative materials able to promote such critical steps.

Point-by-Point Response to Reviewers' Comments

Reviewer #1 (Remarks to the Author):

The authors have revised the paper accordingly and the paper can be publishable.

Response: We thank the reviewer for the recognition of our work.

Reviewer #2 (Remarks to the Author):

I am largely satisfied with the responses and revisions made. The paper is now publishable. I disagree that hydrogen peroxide is kinetically stable at 50 or 70C so this part needs more thought by the authors.

Response: We thank the reviewer for the valuable suggestion. We agree with the reviewer that we should not use the term “*kinetically stable*” in our previous communication (submitted on Nov 25, 2021). What we try to emphasize is that the dissociative activation of H₂O₂ is relatively slow under the methane partial oxidation reaction conditions, as supported by the relatively large dissociation barriers of ~0.56 eV on the Cu dimers. This is comparable to the barrier (0.61 eV) for CH₄ activation on the Cu-dimer sites. We believe it is an advantage of the Cu₂@C₃N₄ catalysts to match the rates of these two steps prior to the recombination to form CH₃OOH or CH₃OH, as indicated by the high gain factor of peroxide utilization and giving rise to high reactivity for partial oxidation of methane.

Reviewer #3 (Remarks to the Author):

As stated in the review of previous version, this is a sound and interesting work dealing with the preparation of Cu-dimers anchored on carbon nitride structure, with significant activity towards methane oxidation to oxygenated products, through thermal and photocatalytic routes. In this revised version, the authors have addressed individual concerns and points related to the catalytic role, stability of dicopper centers and reaction mechanism. In this regard, the authors have conducted longer experiments proving the relative stability of these structures, and have provided more evidence of the proposed mechanism and intrinsic activities, including theoretical calculations. Also, a better explanation of the photocatalytic process has been added, with some experimental insights on the charge transfer from the semiconductor to the catalytic sites, assisted by thermal conditions.

In general, this is a complete work with conclusive evidences of the favored role of catalytic centers towards scission of stable CH₄ molecule and additional steps promoting the activation of the oxidizing species. Therefore, it not only highlights the intrinsic activity of the proposed structures, but also provides ideas for development of alternative materials able to promote such critical steps.

Response: We thank the reviewer for such positive feedback and insightful comments to point out future directions!